# Evaluation of Static Autonomous GNSS Positioning Accuracy Using Single-, Dual-, and Tri-Frequency Smartphones in Forest Canopy Environments

**DOI:** 10.3390/s22031289

**Published:** 2022-02-08

**Authors:** Thomas Purfürst

**Affiliations:** Chair of Forest Operations, University of Freiburg, Werthmannstr. 6, 79085 Freiburg, Germany; thomas.purfuerst@foresteng.uni-freiburg.de; Tel.: +49-761-203-3567

**Keywords:** GNSS, positioning under a forest canopy, multi-frequencies smartphones, Android smartphone, horizontal accuracy, multi-constellation GNSS

## Abstract

Determining the current position in a forest is essential for many applications and is often carried out using smartphones. Modern smartphones now support various GNSS constellations and multi-frequency analyses, which are expected to provide more accurate positioning. This study compares the static autonomous GNSS positioning accuracy under forest conditions of four multi-frequency multi-constellation smartphones as well as six single-frequency smartphones and a geodetic receiver. Measurements were carried out at 15 different study sites under forest canopies, with 24 measurements lasting approximately 10 min each taken for the 11 GNSS receivers. The results indicate that, on average, multi-frequency smartphones can achieve a higher positioning accuracy. However, the accuracy varies greatly between smartphones, even between identical or quasi-identical tested smartphones. Therefore, no accuracy should be generalised depending on the number of usable frequencies or constellations, but each smartphone should be considered separately. The dual-frequency Xiaomi Mi 10 clearly stands out compared with the other smartphone with a DRMS of 4.56 m and has a 34% lower absolute error than the best single-frequency phone.

## 1. Introduction

It is no longer possible to imagine our everyday life and work without being able to determine our current position. With the development and implementation of the global navigation satellite systems (GNSS), an increasing number of applications have been developed based on accurate, independent positioning. Over time, the receivers became smaller, more powerful, and more suitable for everyday use. The potential of position determination in mobile devices such as mobile phones or smartphones was quickly recognised. Therefore, the first mobile phone with a built-in GPS receiver appeared in 1999. It was the Benefone ESC! [1]. Other mobile and smartphones followed. The number of phone models with GPS increased from fewer than 10 in 2005 (e.g., Siemens SXG75 2005 sxg [2]) to more than 200 models by 2017, and 608 support Galileo in 2021 [3,4]. From 2007 to 2020, around 13 billion smartphones were sold. About 47% of the world’s total population owned a smart device in 2020. The integration of GNSS into these devices is nearly 100% [5].

Many applications and business processes rely on positioning, including in non-urban areas such as the forest. The more determinable a position, the more favorable it is for applications. The most straightforward, efficient, quick, and beneficial method is the use of Global Navigation Satellite Systems (GNSS) for satellite-based position determination.

The ways in which GNSS is used in forestry work activities vary between countries [6]. These include, e.g., the localisation of trees and sample plots in forestry [7,8,9,10] and during the felling process [11,12,13], the documentation and navigation of forestry machines [14,15,16,17] and trucks [18], the autonomous control of forestry vehicles [19], and many more Precision Forestry and Industry 4.0 purposes [20].

The possible applications of satellite-based position determination in the forestry sector are diverse and steadily increasing. Therefore, the frequency of use of GNSS technology in current forestry activities is quite high, although it varies between countries [6].

However, GNSS position determination based on trilateration requires a free and unobstructed signal connection to the satellites in order to carry out several precise pseudo-range measurements. With its canopy, which influences, blocks, or reflects the satellite signals to a particular extent, the complex forest environment poses a unique challenge to being able to determine exact positions [21,22,23].

In the scientific literature, many limitations and influence factors of finding a precise GNSS position under the canopy have been identified and examined. These include forest types [8,24,25,26,27], dense canopies [28,29,30,31,32], the water content of leaves and wooden material [33,34], topography [32,35], local atmospheric conditions [36], temperatures below freezing [16,21], seasons [21], wind [21], humidity [27,37], and many more.

The recipient’s environment is of crucial importance. Below the canopy, the Multipath effect can significantly affect the accuracy of the positioning. Multipath occurs when a satellite signal arrives at a receiver’s antenna via more than one different path. Objects such as trees near the receiver antenna can reflect GNSS signals and, thus, create secondary propagation paths. These secondary path signals can interfere with reaching the receiver directly from the satellite. Multipath affects both pseudo-range and carrier-phase measurements [38]. Particularly, thin branches and leaves as strong Multipath reflectors favour the Multipath, reduce the position accuracy, and disturb the signal-to-noise ratio [21,39,40]. In particular, deciduous forests and evergreen forests, where there are tall trees with thick trunks as well as dense crowns, cause high Multipath and poor reception [41]. However, special antennas, for example, polarised or choker ring antennas, which smartphones do not have, can reduce the Multipath effect in shielded areas [23,42].

Multipath error studies of smartphone positioning accuracy in an anechoic chamber in a controlled environment demonstrated that the quality of observations collected in the anechoic chamber is significantly better than those collected in the natural environment [43]. Performance studies of a Xiaomi Mi 8 smartphone related to pseudo Multipath and noise compared to a geodetic receiver found that the smartphones resulted in higher Multipath and lowered C/N${}_{0}$ values than geodetic receivers [44]. Several studies found that Multipath effects can account for about 50% of the horizontal position error observed [42,45,46].

The combination of multi-frequency reception and the relatively new possibility of reading out raw GNSS observation data from smartphones makes it possible to use different positioning variants. Thus, code- and carrier-phase observations can be analysed and thereby Multipath can be minimised.

The positioning accuracy of a GNSS module on a smartphone was typically between 3 and 5 m under good Multipath conditions and over 10 m under harsh Multipath environments [47]. However, measurements of smartphone positioning detected rapid Multipath variations over time [48,49]. Even when positioning was conducted using geodetic class receivers, considerable variability in positioning accuracies occurred due to the Multipath effect [50]. Additionally, the satellite constellations also do not behave in the same way. For example, Galileo measurements have a smaller Multipath error compared to GPS measurements [44].

Several studies have been carried out to investigate the positioning accuracy based on smartphone observations. The accuracies that can be achieved depend crucially on the positioning algorithm used, and results usually reach in the range of metres to decimetres [51,52,53]. A study by Pensyna et al. [54] in 2014 showed that cm solutions are possible in principle, using smartphone antennas, and that the high Multipath sensitivity of smartphones can be overcome. Studies in recent years have shown that centimetre accuracies using smartphones are achievable under good conditions (open sky) [55,56,57], even with the same (Xiaomi Mi 8) or comparable (Huawei P30) smartphones as used in this study [58,59].

A decisive factor in reception quality and position accuracy under forest conditions is the number of available, tracked, and usable satellites. The assumption is that the more satellites in view/track, the better the position accuracy. However, the achievable accuracy also depends on many other factors, such as the distribution of the satellites, the evaluability, and the correction of the delay error variables [21,22,60,61].

In the early research stages, only the American NAVSTAR-GPS (navigation signal timing and ranging—global positioning system) and/or the Russian system GLONASS (globalnaja nawigazionnaja sputnikowaja sistema) could be used. Most modern smartphones now support a multi-constellation of satellite systems and use a combination to determine their position; therefore, they can also use the European GNSS system GALILEO, the Chinese GNSS system BeiDou, the Indian RNSS (Regional Navigation Satellite System) system NavIC (Navigation with Indian Constellation), and the Japanese RNSS system QZSS (Quasi-Zenith Satellite System).

Additionally, the number of satellites that can be used for positioning and the number of frequencies available from the satellites has increased significantly over the last few years, contributing to higher satellite availability, especially in the forest. Figure 1 shows the available operational GNSS/RNSS satellites as a function of years and systems for the years 1978 to 2020.

The number of active satellites available for positioning has tripled since 2002 and has doubled since 2010, with 130 active by the end of 2020. This increase is mainly due to the implementation of the GALILEO and BeiDou systems. A total of 130 active potential satellites was available for positioning the receivers at the beginning of 2021. The current quantitative installation of satellite systems is thus largely complete [62,63,64,65,66,67,68,69]. This high number is particularly noticeable under difficult conditions (e.g., under a forest canopy), as, in most cases, sufficient satellites can now be used in a favourable constellation and thus the DOP (dilution of precision) values are also very good.

GNSS satellite signals may be reflected by the ground and thus influence position determination. When using a ground plane to avoid reflections, higher accuracies are possible under overshadowing [70]. Moreover, an accurate determination of horizontally arranged smartphones can possibly reach other values [53,71]. However, both use cases do not correspond with the practical application of smartphones in the forest, where a smartphone is usually held vertically by the user.

Due to the many position-influencing variables under the forest canopy, the signals’ correction plays a decisive role (differential global navigation satellite system: DGNSS), achieving an accurate position. However, the various correction signals are often not available over a wide area in many areas. The satellite-based augmentation systems (SBASs) are often blocked in forest conditions due to their primarily geostationary satellite altitude and are only available to a limited extent. The same applies to radio- or internet-based DGNSS systems (e.g., SAPOS and BEACON), whose geographical coverage varies greatly and is not comprehensive [16].

In 2018, a chipset from Broadcom BCM47755 that supports two frequencies each for three satellite systems (GPS L1 C/A, L5; QZSS L1, L5; Galileo E1, E5a) was launched; it can also be used on mobile phones [72]. The Xiaomi MI 8 smartphone was the first to use this new multi-frequency chipset, and it was launched in June 2018 [73]. The additional use of the E5a/L5 frequencies is intended to reduce the Multipath effect (i.e., the reflection of satellite signals from trees, etc.) and provides the opportunity to make ionospheric-free linear combinations between observations of two frequencies to eliminate the ionosphere effect, thus improving accuracy [49,74,75]. The reception quality of the Xiaomi Mi 8, as the first dual-frequency smartphone, is particularly often studied [44,51,52,53,71,74,76,77,78,79].

Moreover, a second-generation dual-frequency GNSS chip, the BCM47765, was released by Broadcom in May 2020. It can track new L5 signals, the new BeiDou-3 constellation B2a signal, and signals from the NAVIC constellation. This will increase the availability of dual-frequency signals on three different GNSS constellations instead of two [80,81,82].

Gradually, other important manufacturers also offered dual-frequency chips, including Qualcomm with the Snapdragon X24 LTE modem and HiSilicon with the Kirin 980 system-on-a-chip with high energy efficiency and a form factor for intelligent data processing and FinFET transistor design [82]. Therefore, the number of multi-frequency GNSS smartphones is increasing. Currently, numerous dual-frequency GNSS smartphones are already available. A search resulted in 211 smartphones from 14 different manufacturers [83]. Development is progressing, and there are currently 24 types of smartphones on the market that support three frequencies with BeiDou. In addition, there is progress towards the use of even more free frequencies. There are already five smartphones from Huawei on the market that can use four BeiDou frequencies (B1I + B1B + B1C + B2a) and seven smartphones that can use three Galileo frequencies (E1 + E5a + E2b) [83] (as of 27.12.2021). Therefore, it can be assumed that smartphones’ positioning accuracy in forests will continue to increase, and that there is a need for further research in this field.

This research aims to determine the relatively new multi-frequency multi-constellation smartphones’ absolute GNSS accuracy and compare it with single-frequency multi-constellation smartphones as well as a differential corrected multi-frequency multi-constellation geodetic GNSS receiver (code and phase measurement) under the forest canopy. The position is to be determined based on the position supplied to the user in real time by the Android Location API of the smartphone.

## 2. Materials and Methods

### 2.1. Study Design and Examined Smartphones

This study assesses the accuracy of different single- and multi-frequency GNSS receivers integrated into smartphones under a forest canopy. The experiment examined the accuracy achievable in a static reception status. The priority was the practical evaluation of the internal GNSS receivers’ capabilities under a forest canopy.

A total of 10 smartphones with integrated GNSS modules and one geodetic receiver were used in the tests. The smartphones were selected according to different criteria. In order to obtain a comparison, older models and newer models, as well as models with single and dual-frequency receivers and models that are used in practice (’outdoor/forest mobile phones’), were combined. Table 1 shows an overview of the smartphones and GNSS receivers used in the empirical experiment. It includes the different manufacturers, models, model release years, and the used chips’ hardware (construction) year, from 2015 to 2019. Additional information is provided about the used Android versions and the different API levels from 23 to 29, with all smartphones updated to the highest possible version.

Four out of ten smartphones provide dual-frequency reception (Xiaomi Mi 8, Mi 8 Pro, and Mi 10) with their internal chips, and one smartphone provides triple-frequency reception (Huawei P40). It was often reported from practice that exactly the same smartphone types often lead to very different results. Therefore, a comparison was also made between identical models (Samsung Xcover 4s) and smartphones with the same GNSS chip (Xiaomi Mi 8 and Mi 8 pro).

Figure 2 shows an overview of the different types of smartphones used and their ability to use the different GNSS constellations and frequencies, as well as the possibility to use double-frequency or triple-frequency reception.

### 2.2. Experimental Setting

In order to be able to carry out the measurements simultaneously, a holder was built for the 10 smartphones and the geodetic GNSS receiver. A 1.5 m long aluminium bar was mounted on a tripod. The two-channel Trimble Geo7x GNSS receiver’s external antenna was installed in the middle, precisely above the survey point. Attached to the bar were smartphone holders with an exact distance of 14 cm. The experimental design was aligned consistently east to west, and smartphones were placed with a known defined offset in an east–west orientation. Thus, the position of every smartphone could be derived relative to the known survey point. The smartphones were arranged vertically in the holders.

The tripod was aligned precisely with the centre of the survey point using a plumb bob. The height above the ground varied somewhat due to the settings and the micro-relief, but it was approximately 1 m.

Figure 3 shows the arrangement used in these experiments, including the names of the smartphone models used. The experimental setup was placed exactly over a terrestrially surveyed geodetic survey point where the exact absolute position was known.

The studies took place in Germany in the federal states of Bavaria, Saxony, and Baden-Württemberg. In total, measurements were carried out at 15 different study sites. The tree species, the relief, the degree of canopy closure, and the foliage conditions varied among the study spots. Only sites under a canopy and with official terrestrial survey points of which the absolute accuracy was known were used. Table 2 shows the characteristics of the different study sites.

The locations of the survey points in the forest were acquired from the official survey authorities of the respective federal states. It was a challenge to find points in the forest under a canopy with sufficient accuracy and suitability. In addition, a wide range of real forest structures should be represented. This included both old and young stands in hardwood, softwood, and mixed wood. Only 8% of the potential points were used for the study. According to the surveying offices of the various federal states, the absolute accuracies of the used survey points were less than 5 cm.

The coordinates of the permanently marked survey points were determined in each case as local terrestrial geodetic connection measurements of fixed trigonometric points. The coordinates are available in the official reference system for the federal states of Baden-Württemberg and Bavaria (ETRS89/UTM zone 33N; EPSG: 25832), as well as for the federal state of Saxony (ETRS89/UTM zone 33N (EPSG: 25833). The GNSS coordinates of the smartphones and the geodetic receiver were available in WGS 84 (EPSG: 4326) and were converted into ETRS89/UTM zone 33N (EPSG: 25832) for further processing and analysis by transformation method (ETRS89 to WGS 84 (1); EPSG: 1149) using a Python script (package ESRI arcpy).

The positions in these experiments were provided directly by the internal GNSS receivers of the smartphones and were not externally corrected in any way.

Static positions were measured on the tripod at each study site point for at least 10 min. The smartphones were offline before and during the measurements. In addition, the position service was switched off before the actual measurement. The location detection and the recording function in the app were then switched on in sequence from east to west, and the measurement was carried out. At the end of the measurement, the location option of the smartphones was switched off in the same order.

### 2.3. Data Capturing and Processing Software

All 10 smartphones were equipped with the Android operating system in the latest version available for the respective mobile phone and with all current updates. However, the phones had different versions of the Android system and especially different filters integrated in the analysis of the GNSS raw data, which can affect the analysis of the data.

In this study, GNSS NMEA data provided by the Android Location API are used. As part of the Android 6 (‘Marshmallow’) system, the Location API 23 allows access to the NMEAListner class, providing basic NMEA sentences used in this study. As part of the Android 7 (‘Nougat’) system, the Location API 24+ provides GNSS raw and computed information via Android classes. Therefore, e.g., GNSS clock, receiver time, clock bias, received satellite time, and code- and carrier-phase data are accessible. The Location API 24+ is backwards compatible. Therefore, used functions, such as NMEA sentences, are also available in higher versions [75,84]. With the NMEA data provided by the smartphone’s Android Location API, single point positioning (SPP) can be assumed for internal determination of the position.

The app ‘GPStest’ from the developer ‘barbeauDev’ was used to record the GNSS data. The program code is open source and available on GitHub [85]. For comparison purposes, all smartphones used the same version (3.6.4) of the app ‘GPStest’, and all of the settings of the app were also exactly the same. The app captures the GNSS NMEA (data specification for communication for GNSS/RNSS data by the National Marine Electronics Association) standard data of the mobile phone and prevents the mobile phone from switching off even during long measurements. Furthermore, it is simply structured and has the possibility to display real-time information about the current measurement, especially for the GNSS and SBAS satellites and the different frequencies used. This is essential for a functional check during the capture. Figure 4 shows the Android app ‘GPStest’ used to capture the GNSS data in this study. The app shows the real-time meta and positioning data. Additionally, real-time data about the different satellite identifications, constellations, receiving frequencies (CF), signal-to-noise ratios (C/N${}_{0}$), flags, elevations, and azimuth angles are displayed and stored.

The logged data were further processed as follows:The data from the app ’GPStest’ were stored in the GNSSlog in a text file format on the smartphone. In addition to the header’s metadata, this also contains the navigation message data, raw GNSS measurements data, location fix data, and NMEA data.The data were subsequently read out asynchronously from the individual smartphones and stored in a file system.NMEA data were used to analyse the measured position and satellite data. The app used, ’GPStest’, stores all data in a joint log file containing NMEA data, metadata, navigation data, RAW data, and others. Using a parser, ’GNSS2NMEA’, programmed by the author in Python, the “correct” NMEA data were extracted from the GNSS log file, verified, and saved as a pure NMEA text file. This process was performed for each mobile phone and each recording separately with a batch process.The NMEA data were parsed and written into an MYSQL database. For this purpose, a Python program, ’NMEA2DB’, and database schema created by the author [16,21] were adapted and used to extract the relevant positioning and satellite data from the NMEA file. Some challenges were the different NMEA interpretations, the different NMEA 0183 versions (v2.3; v4.10; v4.11) of the smartphone manufacturers, and the NMEA standard’s different dialects. In particular, the handling of the standard with the different satellite systems and the multi-frequency data was very different. The parser was elaborately and explicitly adapted to the different smartphone models and their NMEA characteristics. The following NMEA 0183 datasets were analysed and the data were stored: (a) RMC: Recommended Minimum Sentence C; (b) GGA: Global Positioning System Fix Data; (c) GNS: GNSS fixed data; (d) GST: GNSS Pseudorange Error Statistics; (e) GSV: Satellites in view); and (f) GSA: GPS DOP and active satellites.The parsing process steps were as follows:(a)Automatic identification of the smartphone types/names. If the type/name was identical (i.e., Samsung XCOVER 4s a and b), this step was manually completed.(b)Extraction of all information from the relevant NMEA sentences.(c)Generation of SQL statements of the parsed data.(d)Execution of the SQL statements to transfer the satellite and position information into a MYSQL GNSS database [16].The position data and the satellite data were separately stored in two different tables (‘pos’ and ‘sat’) that are clearly linked in a one-to-many relationship using measurement-GUID, a receiver (e.g., smartphone), and a UTC timestamp. The tables were fully indexed and query-optimised.

The GNSS data from the Trimble Geo7x were processed separately. Post-processing using the Trimble GPS Pathfinder Office correction service was performed in the office. Afterwards, the corrected position data was loaded into the database. The UTC timestamp can synchronise the positioning data with the smartphone positioning data.

### 2.4. Data Analysis

The data were read from the GNSS database and analysed within the statistical software R (GNU). The satellite data used, the frequency bands, and the carrier-to-noise density ratio C/N0 were extracted and analysed.

Plausibility analysis of the positions provided by the smartphones was carried out. This was necessary because, for example, the Huawei P40, in some cases, first supplies the coordinates of the previous ‘old’ position in the NMEA string after switching on the position data option until a new position is known. In these cases, the ‘old’ data were not considered.

The measured positions were adjusted along the longitude for the distance by which the smartphones deviated in the west–east direction on the holder. This was performed using a view table directly on the database. The adjusted distances are as follows: Mi 10 light: 70 cm; Huawei P40: 56 cm; Huawei P20: 42 cm; Samsung Xcover 4s b: 24 cm; Samsung Xcover 4: 13 cm; Xiaomi Mi 8 pro: −14 cm; Samsung A7: −28 cm; Samsung S5: −42 cm; Samsung Xcover 4s a: −56 cm; and Xiaomi Mi 8: −70 cm.

In this study, only the horizontal error was examined. This was calculated for each individual measurement at 1 s intervals, considering the position error in relation to the respective geodetic survey point. For the evaluation of the absolute error, various mathematical ratios were then derived. In addition to the arithmetic mean and standard deviation, different circular error probable (CEP) values (25, 50, 75, 95, and 99.7), the DRMS (distance root mean square), and the 2DRMS (twice distance root mean square) of the error were derived. The circular error probable (CEP${}_{x}x$) is defined as the radius of a circle centered on the true value that contains xx% of the actual GNSS measurements. It is based on the 2D-error of the GNSS measurement: $\sqrt{{\left(x_{i}-x_{sp}\right)}^{2}+{\left(y_{i}-y_{sp}\right)}^{2}}$, where $x_{sp}$/$y_{sp}$ are the *x*/*y* coordinates of the specific survey point. Equation (Equation 1) shows the calculation of the DRMS.
(1)$$DRMS=\sqrt{\sigma_{x}^{2}+\sigma_{y}^{2}}=\sqrt{\frac{\sum_{i=1}^{N}{\left(x_{i}-x_{sp}\right)}^{2}}{N}+\frac{\sum_{i=1}^{N}{\left(y_{i}-y_{sp}\right)}^{2}}{N}}.$$

A Wilcoxon Rank Sum Test was carried out to test whether the different receivers significantly affected the results. The *p*-value was adjusted using the continuity correction method of ‘holm’ [86]. The significance threshold was set at 0.05.

## 3. Results

Measurements were carried out at 15 different study sites; 24 measurements of approximately 10 min with a recording interval of 1 Hz were made at each position. During the study, a total of 158,150 position data and 4,753,910 satellite data were recorded, parsed, and stored in the GNSS database. The data were collected in different seasons.

### 3.1. Different Signal Reception

The reception quality and the difference between the position accuracies of the various smartphones examined depends strongly on the number, type, and quality of the satellite signals received and used to determine the GNSS position.

Table 3 shows the average number of satellites that were used for position determination per constellation and the frequency bands over the entire data collection session. It is clear that four of the ten tested smartphones also used multiple frequency bands in the field to determine their positions (Xiaomi MI 8, Xiaomi MI 8 Pro, Xiaomi Mi 10 light, Huawei P40). The Huawei P40 used three bands. Older smartphones, such as the Samsung S5 and the Huawei P20, only used two, and the different Samsung XCOVER 4/4s and the Samsung A7 only used three constellation/band combinations. Newer multi-frequencies models, on the other hand, used five (Xiaomi MI 8/8 Pro), seven (Xiaomi Mi 10 light), or even eight (Huawei P40) constellation/band combinations to determine the GNSS position.

The benefit of applying L5/E5a/B2a code pseudoranges may be reliably verified but varies significantly between the different constellations. A careful inspection of the table allows us to conclude that the relative ratio of multi-frequency use by NAVLSTAR GPS satellites is much smaller than that of the other constellation types (BeiDou, Galileo), due to the small number of GPS satellites that provide signals on the L5 frequency band. The average number of acquired GPS satellites on L5 was much lower than that on L1 for all smartphones. The L5 frequency of the GPS NAVSTAR satellites was received by only 26% (Xiaomi Mi 10 light) to 45% (Huawei P40) of the satellites. The lower number of satellites with the L5 band can be explained by its lower availability compared to the availability of the L1 signal. With the B3a/B1l band of the BeiDou constellation, in contrast, the ratio was between 68% and 80%. In the Galileo constellation, except for Xiaomi Mi 8 pro (56%), the E5a band was used for almost every satellite for which E1 was also used. The proportions vary from 92% (Xiaomi Mi 8 pro), to 98.5% (Xiaomi Mi 10 light), to 98.6% (Huawei P40). The average number of acquired GPS satellites on L5 is much lower than that on L1 for all dual-frequency smartphones by an average from 1.6 to 3.4 satellites. Therefore, smartphones that are able to use the second or even third frequency of the BeiDou (B2a, B1) or Galileo (E5a) systems have a potential advantage to eliminate ionosphere interference using the B2a/E5a code pseudoranges, as classical dual-frequency ionosphere-free linear combinations do.

It is particularly remarkable that newer, more frequently used smartphones tended to use significantly more satellites for positioning. Older smartphones (Samsung S5 and Huawei P20) used on average 9.2 or 9.6 active satellites, while modern smartphones (Xiaomi Mi 10 light and Huawei P40) used up to 25.8 or 29.4 satellites for positioning at the same time in the experiment. If one also considers the frequency bands, the number increases to 40.0 and 48.8 bands, respectively. This means that a much higher potential was exploited, and more information was used to determine the position.

Even smartphones that were identical or almost identical in construction differed from each other. However, the two identical single-frequency Xcover 4s smartphones showed very slight, non-significant differences. On the other hand, the Xiaomi MI 8 and Xiaomi MI 8 Pro, which were equipped with the same GNSS chip, differed significantly from each other in some cases. This is particularly evident when using the E5a frequency, of which the Xiaomi MI 8 Pro used the 1.7 (56%) band and the Xiaomi MI 8 the 3.5 (92%) band. The Xiaomi MI 8 used an average of 19 frequency bands to determine its position, whereas the Xiaomi MI 8 Pro used only 14.6 frequency bands under the same recording conditions. This significant difference cannot be explained without a deeper analysis of the hardware and software.

A more detailed representation of the satellites in use and their proportions can be seen in Figure 5. It shows the relative frequency in relation to the satellites in use (not the frequency bands of the received positions).

The relative frequency of the distribution of the satellites for the smartphones is highly diversified. In Figure 5, blue indicates smartphones with multi-frequency capabilities, and red indicates smartphones that can only receive the L1/G1/E1/B1c bands. The recorded time periods, the time stamps, and the recording conditions are identical for all smartphones.

It is evident that the new models, such as the Huawei P40 and the Xiaomi Mi 10 light, use significantly more satellites for positioning, with the former having more than 20 satellites in use over almost the entire study period. This is more than the maximum number for most other smartphones. One explanation for this is the improved hardware, which brings about improved performance with new generations of smartphones and thus a higher number of satellites that can be used. Figure 6a clarifies this relationship using the satellites in use, which were measured in this study in relation to their release dates. In particular, the newer models from 2020 show very high values. This is mainly due to the consistent simultaneous use of all four available constellations. If the different frequency bands are added, this difference becomes even more apparent (Figure 6b).

It is noticeable in Figure 5 that the Xiaomi Mi 8 and the Xiaomi Mi 8 pro both have two-peaked distributions. Both smartphones had partial initialisation problems below the canopy and provided no or only limited position data over more extended periods during the measurements. This was mainly a result of the different canopy cover. If the overshadowing were stronger, then the NAVSTAR GPS would dominate in terms of position detection and Galileo. BeiDou satellites are hardly used anymore, resulting in a number of satellites in use, which are similar to those used by the older models. The Xiaomi Mi 8 pro uses either between 5 and 12 satellites or between 18 and 23 satellites, and the values in between are almost non-existent. The Xiaomi Mi 8, in particular, often had problems with positioning under the canopy, which only provided positions on average around 77% of the time to the other smartphones under the canopy (see Table 4).

In the satellite distribution of the Huawei P 20, a phenomenon occurs that significantly lowers every uneven satellite value. This cannot be explained technically and will undoubtedly affect the smartphone’s evaluation software.

In Figure 7, the skyplots illustrate the mean carrier-to-noise-density ratio C/N${}_{0}$ of satellites in use for the different smartphones as a function of azimuth and elevation. There is a big difference between the satellites used for the same measurements. This is particularly evident in the comparison between the Huawei P40 (Figure 7a) and the Huawei P20 (Figure 7b).

In contrast to high-grade receivers and antennas, there is only a low C/N${}_{0}$ dependence on satellite elevation in smartphone GNSS receivers and a large number of drops in C/N${}_{0}$ that are unexpected for high elevations. These can be seen in Figure 7 for most smartphones. These results indicate that the commonly used elevation-dependent function may not be optimal for GNSS observation weighting and that a C/N${}_{0}$-dependent function, as already proposed [87], would be more appropriate.

It is noticeable that the Xiaomi Mi 10 very often has interruptions in the paths of the active satellites. Therefore, it can be assumed that the smartphone’s evaluation software evaluates and changes the satellites used to determine the position much more differentially and dynamically.

### 3.2. Static Accuracy

Figure 8 illustrates the relative cumulative GNSS accuracy distribution of different smartphones under a forest canopy measured at different known survey points. The multi-frequency smartphones are shown in the various shades of blue, the single-frequency smartphones in the various shades of red, the GNSS reference in olive, and the mean values in green.

It becomes clear that there are significant differences between the smartphones. However, the differences between GNSS receivers vary, so it is difficult to identify the most appropriate mathematical indicator. At this point, the DRMS error will be used for comparability. Table 4 represents the characteristic values of the absolute two-dimensional position deviation of the different smartphones and GNSS receivers used in the empirical experiments.

Under a forest canopy, reception conditions differ significantly in contrast to open areas. Nevertheless, the high-quality corrected Geo 7X (DRMS: 3.26 m) is the most accurate, followed by the Xiaomi Mi 10 light (DRMS: 4.56 m). The others rank from highest to lowest in the following order: Huawei P20 (DRMS: 6.92 m), Samsung A7 (DRMS: 6,94 m), Xiaomi Mi 8 (DRMS: 7.20 m), Huawei P40 (DRMS: 7.22 m), Xcover 4sA (DRMS: 7.73 m), Xcover 4sA (DRMS: 7.98 m), Samsung S5 (DRMS: 8.05 m), Xiaomi Mi 8 Pro (DRMS: 8.55 m), and Xcover 4 (DRMS: 14.59 m). The outdoor smartphone Xcover 4 is far behind with highly inaccurate values.

Based on the measurements of the study, the multi-frequency smartphone receivers (DRMS: 6.99 m) achieve a significantly higher position accuracy under overshadowing than the single-frequency smartphones (DRMS: 9.13 m). It is worth noting that the date of manufacture may also have an influence. A newer release date and thus potentially improved hardware is not a direct prerequisite for improved reception in a forest. Figure 9a illustrates the correlation between the release date and the absolute position error (CEP). The differences (not taking into account the poorly performing Samsung Xcover 4 smartphone) are tiny, and the single-frequency smartphones reach a DRMS value of 7.23 m.

If another index is used, such as the CEP (median), this ratio is put into perspective again, whereby the error difference of multi-frequency smartphones, 4.73 m, is significantly lower than that of the classic single-frequency smartphones, 6.02 m. (see Figure 8 and Figure 10 and Table 4).

A correlation between the absolute position error (CEP) and the number of active satellites is not clear (Figure 9b). The Huawei P20 and the Huawei P40, in particular, clearly demonstrate this, as the P40 uses about three times the number of satellites to determine its position while having similar positioning accuracy. Therefore, other factors greatly influence the precision of the positioning. However, the most accurate smartphone (Xiaomi Mi 10 light) also has a very high number (25.8) of active satellites.

Table 5 shows a Wilcoxon Rank Sum Test to compare the different receivers relating to the absolute position errors. All the accuracy differences between the smartphones are significant, except for the Huawei P40 and the Xiaomi Mi 8 Pro. The identical (Samsung Xcover 4s) or quasi-identical (Xiaomi Mi 8/8 Pro) ones, on the other hand, are significantly different.

## 4. Discussion

This study investigated the static GNSS positioning accuracy of novel multi-frequency multi-constellation smartphones and compared them with single-frequency and geodetic receivers under forest conditions. The acquisition of detailed and spatially accurate positioning data is required in several forest business applications.

It can be assumed that the data provided by multi-constellation, dual-frequency smartphone receivers can reach a higher GNSS accuracy under a forest canopy based on the increase in the signal availability and the elimination of ionospheric error.

Under open sky conditions and special GNSS raw data analyses, such as carrier phase observations, after successful ambiguity fixation, centimeter-level accurate positioning with multi-frequencies smartphones is possible [59,70,78,88].

The current study has focused on the position solution that smartphones deliver directly and in difficult conditions in a forest. Further studies show that the accuracy for point measurements under open area conditions are between two and six times higher compared to point measurements under forest conditions [89]. The reception qualities vary considerably between the different reception conditions (study sites). It is challenging to quantify the environmental conditions under the canopy of a forest that influences reception [21], and available methods, such as free-sky classification, are only conditionally suitable for describing a correlation with position accuracy [16]. Therefore, it is difficult to generalise the results of different GNSS studies under forest conditions. This is particularly true for smartphones as receivers, as even smartphones of the same design deliver different positions to the same reception conditions under a canopy. Even the number of satellites in use alone cannot be considered reliable for estimating position accuracy. Other studies have also come to this conclusion [70].

In static GNSS accuracy studies, the question of how long or how many fixed points should be recorded is always addressed during the experiment setup. The duration of the GNSS recording under a forest canopy has a significant influence on the absolute error and the precision [90,91], and a longer averaging of position values can usually improve it significantly [61,90,92].

In the present study, a recording of at least 10 min with an recording interval of 1 Hz was made at each position. The chosen (optimal) recording time for the experimental determination of static positions has been evaluated differently by different authors and ranges from 2 to 120 min [27,28,92,93,94,95]. However, even 2 min of waiting time for GNSS positioning with a smartphone is too long in practice, as working people usually do not want to wait long for a more accurate position. It is therefore difficult to determine an optimal recording time.

All 15 study sites were considered together in the present study, and this mix was regarded as a representative forest sample. However, there are significant differences in accuracy between the different canopy cover intensities. If the different canopy covering ratios are taken into account [8,70,71], the results can be considered in a more differentiated way. Therefore, further studies should include objective forest stand metrics to compare and correlate different overshadowing situations. A possible method is the Fisheye photos classification algorithm [96], which was used in [16,21,23].

As the reception conditions under large trees are particularly challenging, this is also seen in the considerable variation of error values for all GNSS receivers (Figure 10). There are receiving conditions, especially on the north slope in heavily overshadowed, dense forest, where all types of receivers face challenging conditions for deriving an accurate position. There is a strong significant correlation between canopy density and positional accuracy [21,23], and the accuracy achievable can range from millimeters to tens of meters depending on the operating environment [17].

The dual-frequency smartphones performed best on average across all recordings in this study. It is difficult to determine whether this is due to the multi-frequency technology or progress based on the development of antennas, more sensitive sensors, or improved algorithms. However, there are many differences between individual smartphones. The first multi-frequency smartphone, the Xiaomi Mi 8, achieved a higher accuracy (DRMS: 7.22 m) in this study than the average of the single-frequency smartphones (DRMS: 9.13 m), but individual single-frequency smartphones, such as the Samsung A7 (DRMS: 7.22 m) or the Huawei P20 (DRMS: 7.22 m), achieved a slightly higher accuracy than the Xiaomi Mi 8, as well as the Xiaomi Mi 8 pro (DRMS: 8.55 m). In other studies which compare smartphones under a canopy, the Xiaomi Mi (RMSE: leaf-on: 6.13 m, leaf-off 4.10 m, open arena 2.23 m) outperformed the Huawei P20 lite (RMSE: leaf-on: 8.12 m, leaf-off: 11.44 m, open arena: 3.44 m) [71]. Compared to studies of older smartphone generations under a leafy canopy, widespread accuracies also occur (RSME${}_{xy}$: ZTE Blade: 11.45 m; LG G2 Android 4.4: 9.3 m; LG G2 Android 5.0: 6.74 m; Sony M4 Aqua: 7.48 m; Lenovo Yoga 8: 4.96 m) [89]. However, it must also be considered that the results of the different tests only offer an indication and cannot be directly compared with each other. The studies did not occur under the same conditions, and there are no objective indicators available to evaluate the environmental reception situation.

The multi-frequency Xiaomi Mi 10 clearly stands out compared to single-frequency smartphones. It has a 31% lower absolute error under the CEP and a 34% lower absolute error under the CEP${}_{95}$ compared with the best single-frequency phone. In general, it can be assumed that multi-frequency mobile phones can improve reception under forest conditions. At the same time, it should be noted that this is highly dependent on the capabilities of the individual smartphone and should not be over-generalised. For example, the developers at Xiaomi have significantly improved the reception quality of the Xiaomi Mi 8 and Xiaomi Mi 8 Pro in the Xiaomi Mi 10. In other studies, the Xiaomi Mi 9 showed lower accuracies than the Xiaomi Mi 8 [87]. The overall potential as a combination of available frequencies, hardware, and software will certainly not be exhausted yet. Further research studies on upcoming smartphones should provide more information.

It should be expected that smartphones with identical or very similar design will produce identical results. Therefore, a comparison was made between identical models (Samsung Xcover 3s) and smartphones with the same GNSS chip (Xiaomi Mi 8 and Mi 8 Pro). The two identical Samsung XCover 4s models which were tested, on average, used almost the same number of active satellites for the fixed position (Figure 3 and Figure 6b) but differed slightly in the distribution of satellite frequency (Figure 5) and in the positional accuracy (CEP: 6.02 m vs. 7.22 m; DRMS: 7.73 m vs. 7.98 m). The differences between the two tested Samsung XCover 4s cannot be explained, as they are identical in construction, were bought together, and were treated absolutely the same in terms of software.

The two Xiaomi smartphones (Mi 8 and Mi 8 pro) should have accessed the same satellites and acquired the same position data, as they have the same GNSS chip (Broadcom BCM47755) installed. The similarity between the two smartphones is that they are the only smartphones with a two-top distribution of satellites in use (Figure 5). The positional accuracy was significantly different (CEP: 4.08 m vs. 5.77 m; DRMS: 7.20 m vs. 8.55 m)

However, the differences that occurred confirmed the practitioners’ statements that more significant deviations can occur with absolutely identical phones. A more detailed explanation for this cannot be derived from these studies. These potential differences and the associated fluctuations in position determination must be considered when using and testing these smartphones and should be investigated more closely in the future. Therefore, if possible, two identical smartphones should be constantly tested.

In principle, it should be assumed that newer smartphones with improved hardware and software also achieve higher positioning accuracy. However, the results of this study do not allow a generalised statement to be made (Figure 6a and Figure 9a). Newer hardware and software do not necessarily imply higher accuracy. In [97], it was found that, under open sky conditions, the Samsung S4 and Samsung S5 achieved a higher accuracy than their predecessors, the Samsung S6 and Samsung S7. Other studies found that the Xiaomi Mi 8 had a higher accuracy in GPS- and multi-GNSS solutions than its successor, the Xiaomi Mi 9 [87]. The present study comes to similar conclusions, especially when comparing the Huawei P20 with its successor, the Huawei P40. In the case of the manufacturer Xiaomi, the Xiaomi Mi 10 light, the latest tested mobile phone, is also the one in the current study with the best positioning accuracy under the forest canopy.

The results for the Xiaomi Mi 10 light, in particular, show a more advanced determination of the position could be observed, resulting in higher accuracy. Therefore, the combination of hardware, the version of the operating system, and the application must be considered when determining GNSS measurement accuracy [89]. This leads to the assumption that each hardware and software generation must be checked individually for positioning accuracy both in open sky and under complex conditions such as the forest. For this reason, an international, standardised test method and the regular testing of new smartphones and software variants are advisable.

## 5. Conclusions

Based on several simultaneously measured positions of smartphones, the present study shows that the position accuracy under difficult conditions, such as in the forest, increases due to technical progress. It turns out, the multi-frequency reception and the availability of GNSS raw data of the smartphones, in particular, still have a lot of potential to increase the position accuracy under a canopy.

The current development of smartphone positioning accuracy is not yet complete. Although it is already sufficient for many applications, a higher smartphone location accuracy will result in many benefits and new uses, especially under challenging conditions such as canopy coverage. The trend towards higher accuracy in the mass market has become evident with the introduction of multi-constellation, multi-frequency smartphones. It will undoubtedly continue with, e.g., improvements in the new generation of GNSS chipsets and the smartphone positioning algorithms [49] or other solutions formerly intended for high-end receivers, such as RTK solutions for smartphones [98]. Therefore, there is still a need for development and research in the future.

## Figures and Tables

**Figure 1 sensors-22-01289-f001:**
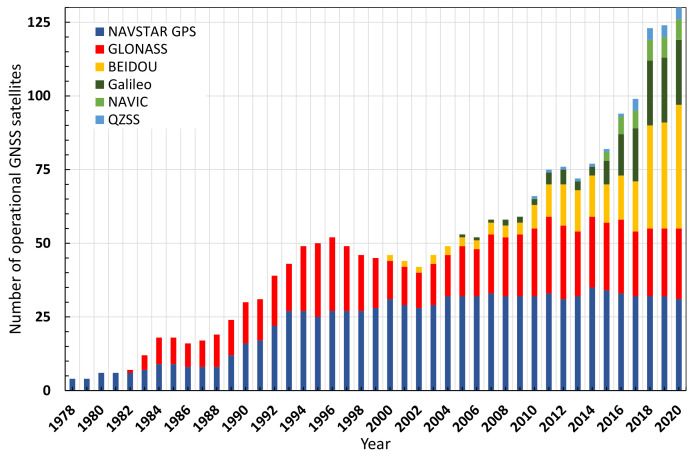
Available operational GNSS/RNSS satellites as a function of years and systems from 1978 to mid-2020 (data sources: [62,63,64,65,66,67,68,69]).

**Figure 2 sensors-22-01289-f002:**
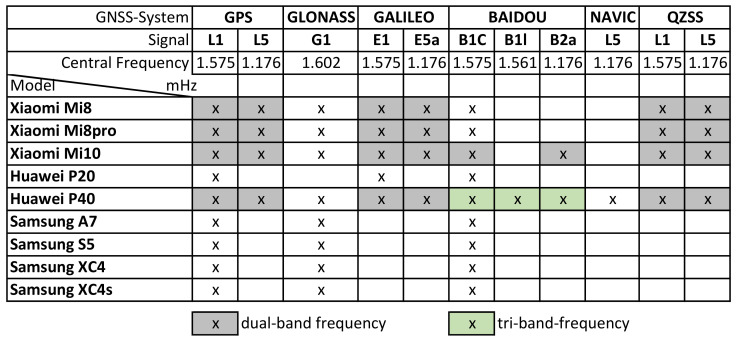
Overview of the different types of smartphones used and their ability to use the different GNSS constellations and frequencies.

**Figure 3 sensors-22-01289-f003:**
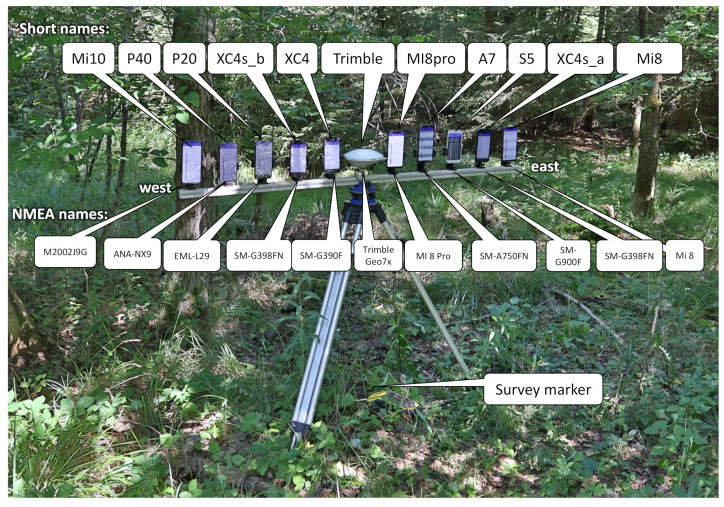
Arrangement of the smartphones and the Trimple GNSS receiver above the survey marker under a forest canopy. The smartphones were placed with a known defined offset in an east–west orientation.

**Figure 4 sensors-22-01289-f004:**
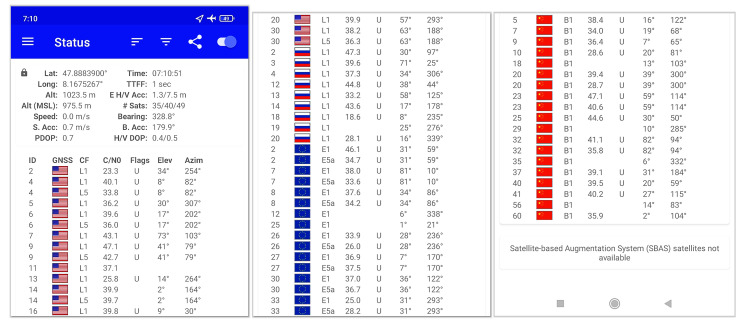
Example smartphone GNSS data from the Android app ’GPStest’ used for capturing and live control of the GNSS data in this study.

**Figure 5 sensors-22-01289-f005:**
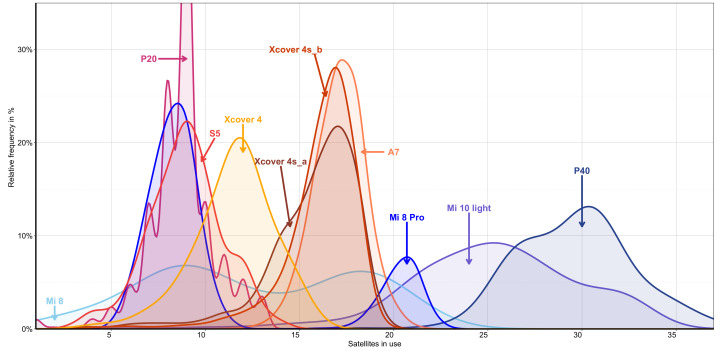
Relative distribution of satellites in use by the smartphones studied. Blue indicates smartphones that have multi-frequency capabilities, and red indicates smartphones that can only receive the L1/G1/E1/B1l bands.

**Figure 6 sensors-22-01289-f006:**
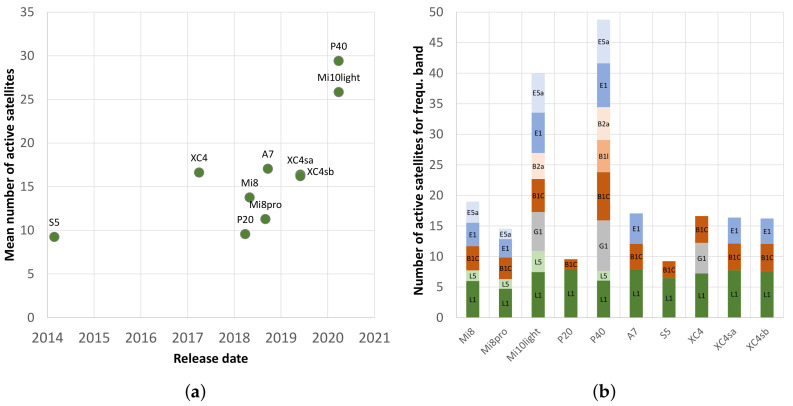
(**a**) Correlation between the release date and the mean number of active satellites used by the smartphones in the whole experiment; (**b**) distribution of active satellites for the different frequency bands (different colors: different frequency bands).

**Figure 7 sensors-22-01289-f007:**
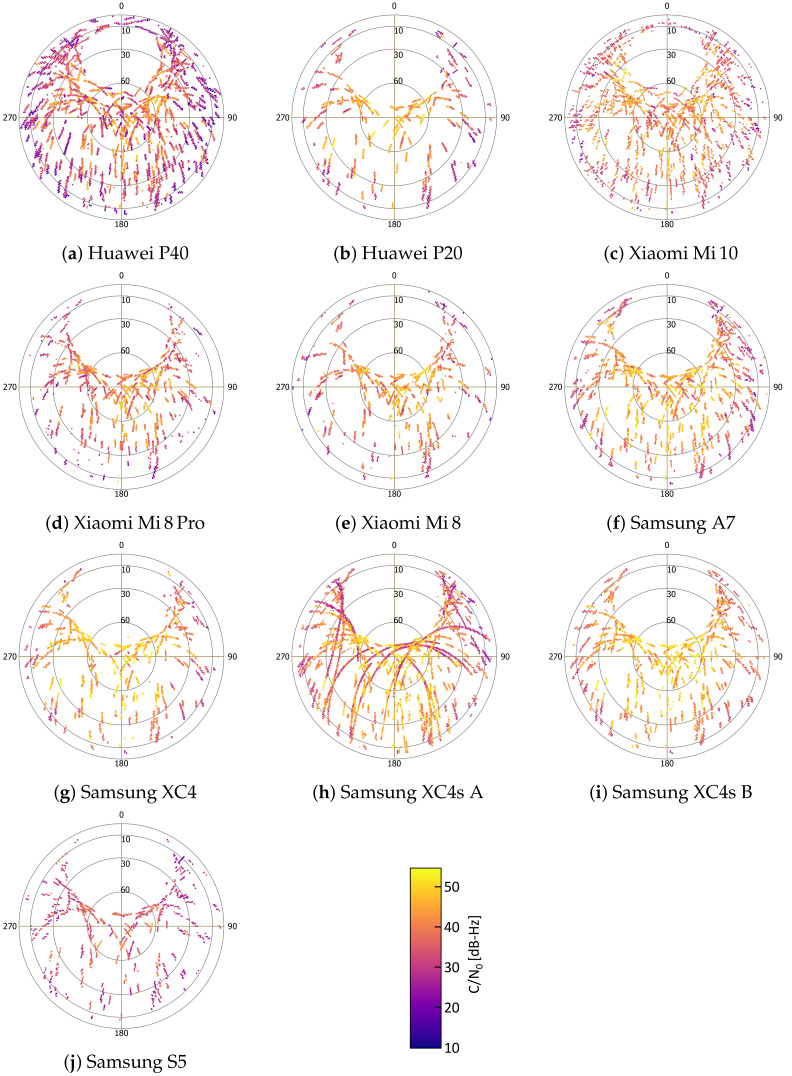
Carrier to noise density ratio C/N${}_{0}$ skyplots of the GNSS signals collected by the different smartphones.

**Figure 8 sensors-22-01289-f008:**
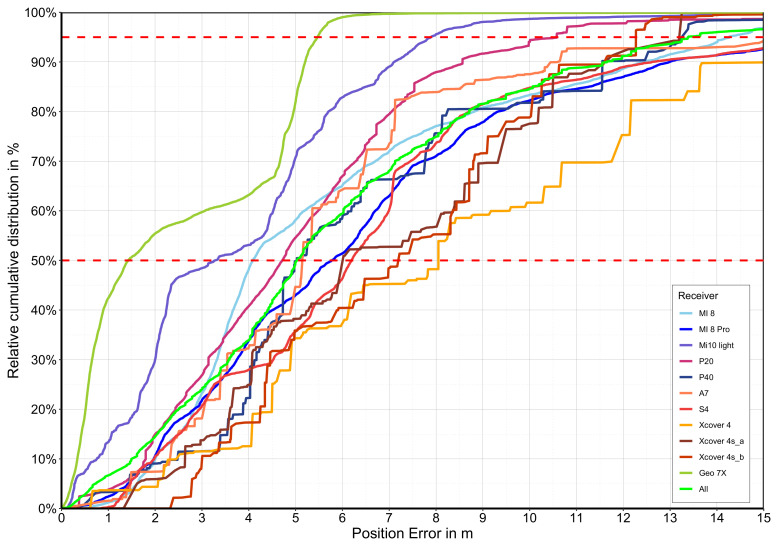
Relative cumulative GNSS accuracy distribution of different smartphones under different forest canopy conditions. Blue indicates smartphones that have multi-frequency capabilities, and red indicates smartphones that can only receive the L1/G1/E1/B1l bands.

**Figure 9 sensors-22-01289-f009:**
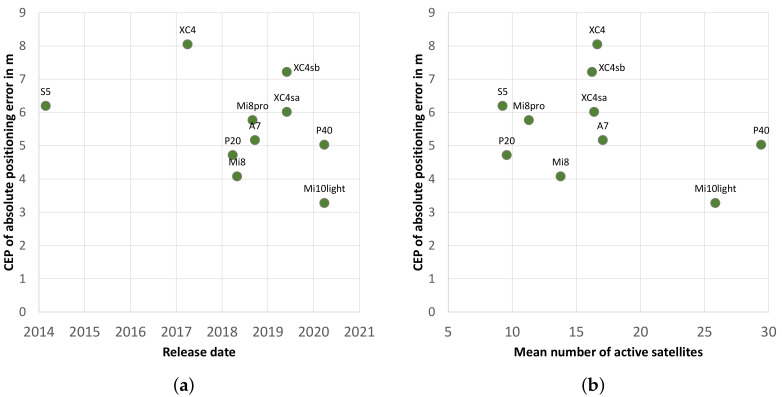
Correlation between the absolute position error (CEP) and (**a**) the release date and (**b**) the mean number of active satellites used by the smartphones used in the experiment.

**Figure 10 sensors-22-01289-f010:**
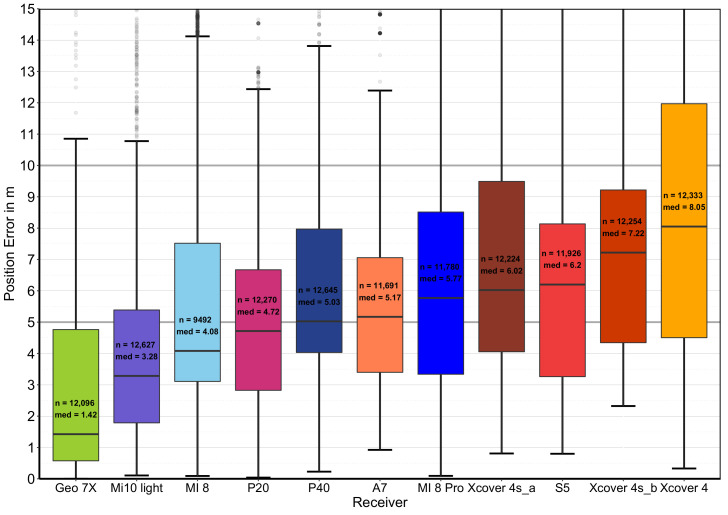
Boxplot of the absolute position error of the different receivers. Blue indicates smartphones that have multi-frequency capabilities, and red indicates smartphones that can only receive the L1/G1/E1/B1l bands.

**Table 1 sensors-22-01289-t001:** Overview of the smartphones and GNSS receivers used in the empirical experiments (${}^{1}$ Number of Frequencies; ${}^{2}$ Release Year/GNSS NMEA Hardware Year/Android Platform/API-Level).

ID	Manufact.	Type	Model	NF ${}^{1}$	RY/HY/P/API ${}^{2}$
Mi8	Xiaomi	Mi 8	Mi 8	2	2018/2018/10/29
Mi8pro	Xiaomi	Mi 8 Pro	MI 8 Pro	2	2018/2018/10/29
Mi10	Xiaomi	Mi 10 light	M2002J9G	2	2020/2019/10/29
P20	Huawei	P20	EML-L29	1	2018/2016/10/29
P40	Huawei	P40	ANA-NX9	3	2020/2018/10/29
S5	Samsung	S5	SM-G900F	1	2014/2013/6/23
A7	Samsung	A7	SM-A750FN	1	2018/2016/10/29
XC4	Samsung	Xcover 4	SM-G390F	1	2017/2015/9/28
XC4s_a	Samsung	Xcover 4s	SM-G398FN	1	2019/2016/10/29
XC4s_b	Samsung	Xcover 4s	SM-G398FN	1	2019/2016/10/29
Trimble	Trimble	Geo7x	TrimbleGeo7x	2	

**Table 2 sensors-22-01289-t002:** Characteristics of the different study sites (SP ID: Official ID of the local survey point; FS: Location in federal state [B: Bavaria; BW: Baden-Württemberg; S: Saxony]).

ID	Latitute	Longtitute	SP ID	FS	Conditions and Main Obstacle for Reception
1	48.0651990	11.5655060	7935 0188	B	Beech dominated mixed forest, 13 m to forest road, closed dense canopy, trees > 15 m
2	48.0502484	11.4392006	7934 0040	B	Beech dominated mixed forest, 6 m to forest road, closed canopy with crown gaps towards the road, trees > 12 m
3	48.0499501	11.4425390	7934 0041	B	Beech dominated mixed forest, 10 m to forest road, closed canopy, trees > 15 m
4	47.9962202	7.7610121	8012 031	BW	Beech dominated deciduous forest, closed canopy, trees > 20 m
5	48.0378690	7.9720488	7913 163 00	BW	Fir dominated mixed forest, northeast slope, closed canopy, trees > 30 m
6	48.0269512	7.9535119	7913 134 00	BW	Beech dominated mixed forest, closed canopy, trees > 25 m
7	47.8880590	8.1546365	8114 034	BW	Spruce, pure stand, medium dense canopy, trees > 25 m, natural rejuvenation > 4 m
8	47.8885525	8.1558580	8114 034 01	BW	Spruce, windthrow area with large open sky areas, trees > 25 m
9	47.8907364	8.1619936	8114 269	BW	Spruce, pure stand, 6 m to forest road, closed canopy with crown gaps towards the road, trees > 25 m
10	47.9650230	7.8463260	8013 025	BW	Beech dominated mixed forest, northward slope, closed canopy, trees > 25 m
11	50.6989621	13.1634072	5244000100	S	Spruce, pure stand, hilltop, canopy with greater gaps, trees > 20
12	50.6971547	13.1657759	5244000101	S	Spruce, pure stand, 2 m to forest road, closed, very dense canopy, crown gaps towards the road, trees > 12 m
13	50.7170483	13.1437748	5244002201	S	Beech dominated mixed forest, closed canopy with small gaps, trees > 20 m
14	50.6900370	13.1400410	5344006100	S	Spruce, pure stand, 4 m to forest road, closed canopy, crown gaps towards the road, trees > 25 m,
15	50.7026740	13.1225790	5244001201	S	Spruce, pure stand, 5 m to forest road, closed canopy with crown gaps towards the road, trees > 25 m

**Table 3 sensors-22-01289-t003:** Number of active satellites used for fixed positions per constellation and frequency bands (FBand) given as an average over the entire data collection session.

Smartphone	GPS		GLONASS	BAIDOU			GALILEO		Sum	
	L1	L5	G1	B1C	B1l	B2a	E1	E5a	Sat’s	FBands
Xiaomi MI 8	6.0	1.8		4.0			3.8	3.5	13.8	19.0
Xiaomi MI 8 Pro	4.7	1.6		3.5			3.0	1.7	11.3	14.6
Xiaomi Mi10 light	7.5	3.4	6.4	5.4		4.3	6.6	6.5	25.8	40.0
Huawei P20	7.9			1.6					9.6	9.6
Huawei P40	6.1	1.6	8.3	7.9	5.3	5.4	7.2	7.1	29.4	48.8
Samsung A7	7.9			4.2			5.0		17.1	17.1
Samsung S5	6.5			2.8					9.2	9.2
Samsung Xcover 4	7.2		5.0	4.4					16.6	16.6
Samsung Xcover 4s A	7.7			4.4			4.3		16.4	16.4
Samsung Xcover 4s B	7.6			4.5			4.1		16.2	16.2

**Table 4 sensors-22-01289-t004:** Characteristic values of the absolute two-dimensional position deviation of the different smartphones and GNSS receivers used in the empirical experiments. (MF: multi-frequency; $\bar{x}$: arithmetic mean; SD: standard deviation; $CEP_{xx}$: circular error probable of xx%; CEP: circular error probable of 50%; DRMS: distance root mean square; 2DRMS: twice distance root mean square, n: number of measurements).

Receivers	MF	$\bar{x}$	SD	CEP${}_{25}$	CEP	CEP${}_{75}$	CEP${}_{95}$	CEP${}_{99.7}$	DRMS	2DRMS	n
		(m)	(m)	(m)	(m)	(m)	(m)	(m)	(m)	(m)	
Geo 7X	Yes	2.49	2.11	0.57	1.42	4.76	5.46	6.97	3.26	6.52	12,096
P20	No	5.22	4.54	2.82	4.72	6.67	10.58	37.64	6.92	13.83	12,270
P40	Yes	6.28	3.58	4.03	5.03	7.97	13.38	16.00	7.22	14.45	12,645
MI 8	Yes	5.85	4.19	3.11	4.08	7.52	14.27	22.53	7.20	14.39	9492
MI 8 Pro	Yes	6.75	5.26	3.34	5.77	8.51	16.34	30.46	8.55	17.10	11,780
Mi10 light	Yes	3.73	2.62	1.79	3.28	5.39	7.89	16.10	4.56	9.13	12,627
Xcover 4	No	10.44	10.19	4.50	8.05	11.97	41.99	47.25	14.59	29.17	12,333
Xcover 4s_a	No	6.90	3.48	4.06	6.02	9.49	13.25	13.25	7.73	15.46	12,224
Xcover 4s_b	No	7.20	3.44	4.34	7.22	9.22	12.27	19.72	7.98	15.96	12,254
A7	No	5.86	3.70	3.40	5.17	7.06	15.80	17.74	6.94	13.87	11,691
S5	No	6.73	4.41	3.26	6.20	8.14	16.72	22.55	8.05	16.09	11,926
MultiF No	No	7.07	5.77	3.75	6.02	8.79	13.64	41.99	9.13	18.25	72,698
MultiF Yes	Yes	5.62	4.16	2.79	4.73	7.32	13.38	24.04	6.99	13.98	46,544
All		6.14	5.17	3.08	5.03	8.03	13.38	41.99	8.03	16.05	131,338

**Table 5 sensors-22-01289-t005:** Wilcoxon Rank Sum Test with continuity correction and ‘holm’ correction between the absolute positioning errors of the studied smartphones (*p*-values, α = 0.05).

	Geo 7X	Mi 8	MI 8P	MI 10	P20	P40	A7	S5	XC4sa	XC4sb
Xiaomi Mi 8	0.00									
Xiaomi Mi 8 Pro	0.00	0.00								
Xiaomi Mi 10 light	0.00	0.00	0.00							
Huawei P20	0.00	0.00	0.00	0.00						
Huawei P40	0.00	0.00	0.88	0.00	0.00					
Samsung A7	0.00	0.00	0.00	0.00	0.00	0.00				
Samsung S5	0.00	0.00	0.00	0.00	0.00	0.00	0.00			
Samsung Xcover 4s_a	0.00	0.00	0.00	0.00	0.00	0.00	0.00	0.00		
Samsung Xcover 4s_b	0.00	0.00	0.00	0.00	0.00	0.00	0.00	0.00	0.00	
Samsung Xcover 4	0.00	0.00	0.00	0.00	0.00	0.00	0.00	0.00	0.00	0.00

## Data Availability

The data that support the findings of this study are available from the corresponding author upon reasonable request.

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
