# Peer review of "Evaluation of Static Autonomous GNSS Positioning Accuracy Using Single-, Dual-, and Tri-Frequency Smartphones in Forest Canopy Environments"

_sensors, 2022, doi:10.3390/s22031289_

Round 1

Reviewer 1 Report

The paper deals with an interesting subject of accuracy of smartphone GNSS receivers under conditions of forest canopy. Even if the idea isn’t very novel, it brings some interesting insight and it is definitely a valuable contribution to this area of research. The experiment is well designed although the methodology needs some minor clarification. The only major flaw, which I see with the respect to the specific conditions in forest, is that the reader gets very little information (almost none) about the conditions where the measurements were conducted. Therefore I strongly recommend to add: 1. table with basic characteristics of the individual sites (perhaps age/density, dominant species, average diameter and height, some terrain classification). It would be appropriate in methodology or, in case of higher extent, as an appendix. 2. table or graph with some accuracy metrics (perhaps DRMS) according to particular sites. I totally agree with the author that generalization is very hardly possible with regard to devices and also with regard to very variable forest conditions. But I think that dealing with the forest just as an “average” forest impoverishes the study, even if some stronger relation between some forest stand metrics and the accuracy is unlikely to be found.

The majority of my more specific suggestions and comments is in the attached PDF file in the form of commentaries.

After considering the suggestions, I think that the manuscript could be published in the Sensors journal and it will be a valuable contribution to the research on the usability of smartphone GNSS receivers under conditions of forest canopy.

Author Response

Reviewer #1:

               Direct comments

The paper deals with an interesting subject of accuracy of smartphone GNSS receivers under conditions of forest canopy. Even if the idea isn’t very novel, it brings some interesting insight and it is definitely a valuable contribution to this area of research. The experiment is well designed although the methodology needs some minor clarification. The only major flaw, which I see with the respect to the specific conditions in forest, is that the reader gets very little information (almost none) about the conditions where the measurements were conducted. Therefore I strongly recommend to add:

  1. table with basic characteristics of the individual sites (perhaps age/density, dominant species, average diameter and height, some terrain classification). It would be appropriate in methodology or, in case of higher extent, as an appendix.

Author: Table added as new table 3

  1. table or graph with some accuracy metrics (perhaps DRMS) according to particular sites. I totally agree with the author that generalisation is very hardly possible with regard to devices and also with regard to very variable forest conditions. But I think that dealing with the forest just as an “average” forest impoverishes the study, even if some stronger relation between some forest stand metrics and the accuracy is unlikely to be found.

Author: Actually, you are right and this information would be beneficial to the work. However, I have deliberately decided against it. The original plan was to use an objective measure of overshadowing. For this purpose, fisheye photos of the canopy were taken at all locations. In previous studies, I did convert these Fisheye photos to greyscale and determined the percentage values of the sky openness using the algorithm of Schwalbe et al. (2009) . Unfortunately, due to a camera error, the quality of the images taken was faulty and the derivation of this index was no longer possible. A subsequent survey would mean that at least 700 km or 400 km (oneway) would have to be travelled just for these pictures. This can also only be done sensibly based on a leafed condition. Therefore, this objective indicator was (unfortunately) not used in the studies and the results can only be generalised. The table would only show the variability between the different sites. Furthermore, this information would not directly contribute to the aim of this research (although this correlation would be scientifically very interesting). If you insist I can insert such a table and will do so. I would prefer to omit them. I have added another paragraph to this topic in the discussion.

The majority of my more specific suggestions and comments is in the attached PDF file in the form of commentaries.

After considering the suggestions, I think that the manuscript could be published in the Sensors journal and it will be a valuable contribution to the research on the usability of smartphone GNSS receivers under conditions of forest canopy.

PDF comments

L0: I suggest adding "static autonomous GNSS positioning" here, or at least in the abstract, as the "static" alone could suggest differential positioning.

Author: changed

L4: the same as in the topic

Author: changed

L6: simultaneously? On all sites in the same time? I would drop this.

Author: changed/dropped

L19: systems

Author: changed

L29: I would change it to "including"

Author: changed

L45: "blocks" perhaps

Author: changed

L46: I would change it to "to determining the exact position"

Author: changed

L54: "tracked"?

Author: changed

L54: I understand the sentence but I'm somehow not content with it. Perhaps "The tendency is that the more is the better"

Author: changed to ’The tendency is that the more satellites in view/track, the better the position accuracy.’

L63: the abbreviation is not introduced, please, do it

Author: added

L72 BeiDou. Please check this throughout the manuscript as it is frequent.

Author: All changed to BeiDou

L74: Is the new paragraph necessary?

Author: no; changed

L82: blocked

Author: changed

L117: I don't think that the Geo7x is currently a high end receiver. If I see correctly, author use and external antenna with this handheld receiver. So, please, clarify this.

Author: changed to ‘high end  geodetic receiver’

L130: I agree, but don't you have some reference for this by a chance?

Author: Adding a reference is not easy. I received the information after a presentation on the positional accuracy of smartphones as feedback from several participants of the KWF Working Committee on Forest Information Management. The IT department heads of the German state forestry enterprises/administrations are mainly represented in this committee and are thus the contact persons for the more than 10,000 smartphones used in the forest in these enterprises. Therefore, this is verbal communication, and I would omit the citation and leave the statement generalised.

L154: Please, add a table with basic characteristics of the sites. Here, or as an appendix.

Author: Table added as new table 3

L155: Which, why? I understand if there were just practical reasons (e.g. availability), but please clarify this.

Author: This side sentence that measurements were carried out several times at individual sites was only intended to indicate that many more measurements were carried out, but in the end only the measurements where everything worked out and the 11 devices actually recorded coordinates were used. I have deleted the subordinate clause. 

L165: Could you please add more info about the reference. E.g. method used (if known) or the coordinate system (crucial for transformations, when dealing with GNSS)

Author: I have added a paragraph: ‘The coordinates of the permanently marked survey points were determined in each case as local terrestrial geodetic connection measurements of fixed trigonometric points. The coordinates are available in the official reference system for the federal states of Baden-Württemberg and Bavaria (ETRS89 / UTM zone 33N; EPSG: 25832) as well as for the federal state of Saxony (ETRS89 / UTM zone 33N (EPSG: 25833). The GNSS coordinates of the smartphones and the geodetic receiver were available in WGS 84 (EPSG: 4326) and were converted into ETRS89 / UTM zone 33N (EPSG: 25832) for further processing and analysis by transformation method (ETRS89 to WGS 84 (1); EPSG: 1149) using a Python script (package ESRI arcpy).’

L166ff: This would probably be more appropriate in Discussion.

Author: The paragraph has to be rearranged in the discussion

L182: on first look, this reads as if the location detection was off during the measurements. Please consider modifying these two sentences.

Author: changed to ‘The smartphones were offline before and during the measurements. In addition, the position service was switched off before the actual measurement.’

S6 Footnote: Isn't it possible to add these as standard references?

Author: changed: The footnotes have been linked or included in the text for reference.

L261ff: I suggest adding at least the two most important metrics (DRMS, CEP50) as formulas here Author: I am always a bit divided about writing standard formulas in paper. The CEP50 corresponds to the median of all 2d errors. The median and its calculation should be known. The same applies to other quartiles (other CEPs). I have added the formula for the DRMS and specified the calculation of the CEP in the text.

L287ff: This is to a high extent the same as the paragraph in lines 302-311. Please consider joining these two sections.

Author: The paragraphs are rearranged and joint.

L340: I don't understand this.

Author: The sentence was reworded.

L341: canopy cover?

Author: changed to ‘canopy cover’

L359: Couldn't this be caused by the orientation (vertical) of the smartphones and the principle of their antenna? Just a future subject perhaps, no need to answer it now.

Author: Interesting point. Several authors examined the smartphones horizontal (I mentioned it in the text [ref: 59-61]). This is an interesting point for research. It would be possible to use two identical smartphones and compare them and gradually change the orientation of one of them.

L369: I don't see different survey points in Fig. 7. I suggest adding a figure or a table with that regard.

Author: I did answer this in the above note

L380: differential corrections were used with this receiver? Please clarify this in the methodology.

Author: You are right, I forgot to mention it in the methodology. I have added a paragraph there.

L401ff: This should be focused in Discussion.

Author: The paragraphs are rearranged and extended in the discussions

L413ff: This could be hard to understand even if in fact it's OK. In first sentence you use "correlation" while "difference" in conjuction with "non-significant". How about to simplyfy this to something like: "All the accuracy differences between the smartphones are significant, except for P40 and Mi8pro."

Author: changed

L418: Overall, author just lists studies with similar focus, rather than confronting them on the numerical basis. Please do this in the few cases where it is possible.

Author: Several numerical results from literature added in the discussion

L426: Is this true for autonomous smartphone measurements?

Author: No. You are right. In this study I just using the NMEA data provided from the Location API. changed.

L446: This is just repeating the Results, please, consider droping such parts when not confronted with particular results of other authors.

Author: à the paragraph was changed, repeating results were deleted, and general, the discussion was more filled with confronted numerical data

L473: Why specifically these two?

Author: The idea was to present and follow the development of a smartphone type. However, it would be better to keep this general, so I removed the two names.

L490ff: This could be the same paragraph.

Author: A mistake happened here. A sentence was accidentally commented out by a note during the language check and I didn't notice it. That's why there is now one more sentence here and the paragraph makes sense.

L510ff: Consider dropping one "However"

Author: changed

L518: however

Author: changed

L529ff: I would separate this as a Conclusion, perhaps with some expansion.

Author: changed

Ref74: The references n. 74 and 77 are actually the same. Please, check this throughout the manuscript.

Author: changed

Reviewer 2 Report

The paper has an interestingly designed study that is done with multiple devices, on several different sites. The analyses of the data obtained seem correct and the conclusions drawn, although sometimes surprising, are valid.However, it has deficiencies that must be corrected for the paper to be accepted.

The author mentions a multitude of studies on GNSS measurements in forested areas. It would be worthwhile to briefly describe their results, so that one could refer to them by comparing the results obtained by the author. In this context, it would also be necessary to justify what is novelty about the results obtained by the author in relation to the previous research.

In the introduction, very little attention is paid to the phenomenon of multipath, which is a very important factor limiting the accuracy of positioning in the vicinity of trees or tall buildings. I believe that this topic should be a bit more elaborate.

It should be also described a bit more why, in general, measurements on two frequencies give much better results. I mean the way of dealing with ionospheric delay - it would be worth at least briefly describing the algorithms that distinguish these two approaches.

As a GNSS expert and a layman in the field of forestry, I would be interested to read whether the GNSS static measurement has any use in forestry or is just a synthetic test, comparing the positioning capabilities of different phones for research purposes. I presume that for measurements used in forestry, kinematic measurements are primarily used but I could be wrong here.

Other remarks:

line 45: Referring to code GNSS measurements we should use the term pseudo-range measurement, not distance measurement.

line 66-69:It is worth mentioning that apart from the development of systems, in the form of a greater number of satellites, the transmission of more signals on different frequencies has been introduced. This also contributed to an increase in positioning accuracy.

line 72: Beidou. The name of the Chinese system is once called Beidou and once Baidou. Please use the correct name throughout the work.

line 82: "In forests..." This sentence seems stylistically incorrect. It needs to be redrafted.

line 85. "... and is not comprehensive" This sentence is incomprehensible. Currently, in all highly developed countries, GBAS stations are densely distributed, allowing them to be used basically everywhere. Of course, this applies to geodetic measurements and receivers adapted to receive such corrections.

line 91: Multipath

line 109: Just to be sure: did Author use differential positioning (so based only on code measurements) or relative positioning (code+phase measurements)? The results obtained would indicate a code-based differential measurement, but perhaps the not-so-high accuracy for a geodetic receiver is due to the large tree canopy.

line 113-114: "multi-frequency possibility" is not a very good wording. Maybe better would be something like: The study assesses the accuracy of different single- and multi-frequency GNSS receivers integrated into smartphones collecting observations under a forest canopy.

line 117: geodetic receiver

line 127: "Four out of ten..."

line 306-307: The lower number of satellites with L5 band can be explained by lower availability compared to the availability of L1 signal. But what about the difference between smartphones?

Fig 4: I do not fully understand the % frequency ratio. I did not find this element well described in the text.

line 360-362: This might have a strong correlation with the multipath effect of which there is not much information about in the text.

line 425: Ambiguity resolution applies to GNSS phase measurements, which are not studied in this paper!

line 427: The author so far in the paper does not mention much about phase measurements. The measurements made in the paper are most likely code measurements. So the ambiguity resolution is not applicable here! The whole paragraph could be found in the introduction, in the review of similar studies but should be removed here.

line 433: "Therefore..." This sentence is unnecessary and follows directly from the previous one

line 434: I can't quite agree with that. Observation conditions can be described in many ways: number of satellites observed, PDOP, signal to noise ratio, etc. which Author even used in his paper.

line 475-489: These 2 paragraphs seem to be interjected a bit out of order. They separate 2 paragraphs summarizing the results obtained in the study and contain information that definitely fits more into the introduction than the summary.

line 521: This sentence is not clear. "differential determination of the position" brings to mind DGNSS measurements, which the Author does not seem to have in mind in this case.

Author Response

Reviewer #2:

Direct comments

The paper has an interestingly designed study that is done with multiple devices, on several different sites. The analyses of the data obtained seem correct and the conclusions drawn, although sometimes surprising, are valid.However, it has deficiencies that must be corrected for the paper to be accepted.

The author mentions a multitude of studies on GNSS measurements in forested areas. It would be worthwhile to briefly describe their results, so that one could refer to them by comparing the results obtained by the author. In this context, it would also be necessary to justify what is novelty about the results obtained by the author in relation to the previous research.

Author: à I changed several sentences/paragraphs following your suggestion and did address what is novel in previous research

In the introduction, very little attention is paid to the phenomenon of multipath, which is a very important factor limiting the accuracy of positioning in the vicinity of trees or tall buildings. I believe that this topic should be a bit more elaborate.

Author: I added two paragraphs about multipath in the introductions

It should be also described a bit more why, in general, measurements on two frequencies give much better results. I mean the way of dealing with ionospheric delay - it would be worth at least briefly describing the algorithms that distinguish these two approaches.

Author: changed in the introduction

As a GNSS expert and a layman in the field of forestry, I would be interested to read whether the GNSS static measurement has any use in forestry or is just a synthetic test, comparing the positioning capabilities of different phones for research purposes. I presume that for measurements used in forestry, kinematic measurements are primarily used but I could be wrong here.

Author: In forestry, there are many applications for both static and kinematic position determination. Static position determination plays a particularly important role in the localization of standing trees, damaged trees (e.g. bark beetles), wood stacks, other things and the use of inventory points. But of course, there are also many applications for kinematic position determination, such as machine travel documentation, product tracking, navigation, etc. However, for the kinematic accuracy studies, other experimental set-ups are needed. We are currently constructing something similar to a railway caring a GNSS receiver for such kinematic experiments. 

Other remarks:

line 45: Referring to code GNSS measurements we should use the term pseudo-range measurement, not distance measurement.

Author: Your are right, changed

line 66-69:It is worth mentioning that apart from the development of systems, in the form of a greater number of satellites, the transmission of more signals on different frequencies has been introduced. This also contributed to an increase in positioning accuracy.

Author: information added

line 72: Beidou. The name of the Chinese system is once called Beidou and once Baidou. Please use the correct name throughout the work.

Author: All changed to BeiDou

line 82: "In forests..." This sentence seems stylistically incorrect. It needs to be redrafted.

Author: The sentence was stylistic incorrect and has been reworded

line 85. "... and is not comprehensive" This sentence is incomprehensible. Currently, in all highly developed countries, GBAS stations are densely distributed, allowing them to be used basically everywhere. Of course, this applies to geodetic measurements and receivers adapted to receive such corrections.

Author: I agree partly. The stations are densely distributed but usually in rural areas and not usable in all forest regions. Anyhow, for the measurement with smartphones they have no relevance and therefore I have taken them out of the sentence.

line 91: Multipath

Author: changed

line 109: Just to be sure: did Author use differential positioning (so based only on code measurements) or relative positioning (code+phase measurements)? The results obtained would indicate a code-based differential measurement, but perhaps the not-so-high accuracy for a geodetic receiver is due to the large tree canopy.

Author: The geodetic receiver used a code+phase measurements. The problem is really the sometimes very dense multi-layered crowns, where the signals are so strongly disturbed/reflected. I have added addressed it in the text.

line 113-114: "multi-frequency possibility" is not a very good wording. Maybe better would be something like: The study assesses the accuracy of different single- and multi-frequency GNSS receivers integrated into smartphones collecting observations under a forest canopy.

Author: You are right the word is very constructed. Replaced and changed

line 117: geodetic receiver

Author: changed

line 127: "Four out of ten..."

Author: changed

line 306-307: The lower number of satellites with L5 band can be explained by lower availability compared to the availability of L1 signal. But what about the difference between smartphones?

Author: I have changed the sentence.

Fig 4: I do not fully understand the % frequency ratio. I did not find this element well described in the text.

Author: I changed the figure axle description  to “relative frequency in %” and write more in the text.

line 360-362: This might have a strong correlation with the multipath effect of which there is not much information about in the text.

Author: I added two paragraphs about multipath in the introductions. Additional in the discussion.

line 425: Ambiguity resolution applies to GNSS phase measurements, which are not studied in this paper!

Author: You are right. In this study I just using the NMEA data provided from the Location API. changed.

line 427: The author so far in the paper does not mention much about phase measurements. The measurements made in the paper are most likely code measurements. So the ambiguity resolution is not applicable here! The whole paragraph could be found in the introduction, in the review of similar studies but should be removed here.

Author: I would like to leave the sentence in the discussion. In the discussion it should be pointed out that under other circumstances (not in this study) it is also possible to use other solutions for position determination using raw data and thus possibly achieve better results with smartphones. However, I have adjusted the sentence somewhat.

line 433: "Therefore..." This sentence is unnecessary and follows directly from the previous one

Author: deledet

line 434: I can't quite agree with that. Observation conditions can be described in many ways: number of satellites observed, PDOP, signal to noise ratio, etc. which Author even used in his paper.

Author: There is a misunderstanding here. I do not mean the dependent variables such as PDOP ect. but the objective description of the environmental conditions as independent variables.  I have clarified the sentence

line 475-489: These 2 paragraphs seem to be interjected a bit out of order. They separate 2 paragraphs summarising the results obtained in the study and contain information that definitely fits more into the introduction than the summary.

Author: The paragraphs have to be rearranged in the introduction

line 521: This sentence is not clear. "differential determination of the position" brings to mind DGNSS measurements, which the Author does not seem to have in mind in this case.

Author: text linguistically adapted

Additional PDF comments

L108: traditional

Author: changed

L127: geodetic receiver

Author: changed

L147: the ground

Author: changed

Table 1: Baidou

Author: All changed to BeiDou

L288: BaiDou

Author: All changed to BeiDou

L292: BaiDou

Author: All changed to BeiDou

Table3: receivers used

Author: changed

L478: Baidou

Author: All changed to BeiDou

L480: Baidou

Author: All changed to BeiDou

L490: designs

Author: changed

L501: quite

Author: changed

L511: However

Author: changed

Round 2

Reviewer 2 Report

The author has scrupulously taken into account all comments to the first version of the work. I therefore recommend that the paper be accepted for publication.

Author Response

Reviewer

This article is interesting, the reviewer comments have been satisfactorily answered, but there are some explicit clarifications that need to be made for me to accept it for publication.

1) For instance, the authors only cite one article that show that cm-level accuracy is possible under good conditions at line 74-76, and this is even in a German article and not an international peer-reviewed journal article. There are at least one much earlier study that has shown that cm-level positioning is possible using smartphone antennas, published in a peer-reviewed international conference, for instance,

Pesyna, K. M., Heath, R. W., & Humphreys, T. E. (2014). Centimeter positioning with a smartphone-quality GNSS antenna. In Proceedings of the 27th International Technical Meeting of the Satellite Division of The Institute of Navigation (ION GNSS+ 2014) (pp. 1568-1577).

And there is also very recent peer-reviewed journal that further shows that cm-level positioning accuracy is possible using the smartphone internal antennas,

Yong, C. Z., Odolinski, R., Zaminpardaz, S., Moore, M., Rubinov, E., Er, J., & Denham, M. (2021). Instantaneous, Dual-Frequency, Multi-GNSS Precise RTK Positioning Using Google Pixel 4 and Samsung Galaxy S20 Smartphones for Zero and Short Baselines. Sensors, 21(24), 8318. doi: 10.3390/s21248318

In Yong et al. (2021) above, this was shown, for the first time, to be possible even in instantaneous (single-epoch) real-time kinematic (RTK) mode, whereas all other studies have used dynamic models in a Kalman filter (multi-epoch mode) to obtain cm-level positioning accuracy.

Author: I have taken your valuable comments into account and have rearranged the section on literature and added more literature. I have also moved the paragraph in the article slightly to the back, where it fits better. 

2) The authors do not address why they do not do relative positioning, such as RTK positioning. Is it single point positioning (SPP) you are using? Please clarify explicitly which functional model that has been used.

Author: The aim of the study was not to increase the accuracy of the smartphones by applying or developing additional "external" PP analysis methods, but to determine how high the accuracy of the integrated methods/algorithms is and thus what accuracy a user of the mass-market GNSS device smartphone can actually expect standalone under these conditions. Therefore, no raw data but the NMEA data provided by the Location API was evaluated, and neither relative position determination nor external phase evaluation was carried out. Despite an intensive search, I have not been able to determine whether the internal position data really only determines the position via the code or the phase. This could also differ between smartphones and versions. Every PPP-based evaluation I have found works with raw data. I would be grateful for a hint/literature. Strictly, the positioning method is not relevant for the measurements in this study, but it is for understanding the data. The use of raw data is certainly the next step that should be studied. I have clarified the methodology and reasoning for this in several places in the text.

3) There are some strange ordering of sections. First we have "1 Introduction", then this follows by "2.2 Experimental setting". Where is Section 2 and Section 2.1? Please correct.

Author: The ordering of sections are correct. It was an editing problem. The version I did submit was complete and well ordered. I think the MDPI office did a track-change with latexdiff and while this compiling process nearly one page incl. Figure 1 was lost. Therefore, I didn't submit a latexdiff track-change at the first revision but a PDF-Compare-track-change, which was lost. Nothing changed here because it should be OK.

4) Table 5 does not have any units, is this meters for all the statistical measures? If so please specify this in the caption.

Author: changed in Table
